# Effect of Silymarin Supplementation on Physical Performance, Muscle and Myocardium Histological Changes, Bodyweight, and Food Consumption in Rats Subjected to Regular Exercise Training

**DOI:** 10.3390/ijms21207724

**Published:** 2020-10-19

**Authors:** Nancy Vargas-Mendoza, Marcelo Ángeles-Valencia, Eduardo Osiris Madrigal-Santillán, Mauricio Morales-Martínez, Judith Margarita Tirado-Lule, Arturo Solano-Urrusquieta, Eduardo Madrigal-Bujaidar, Isela Álvarez-González, Tomás Fregoso-Aguilar, Ángel Morales-González, José A. Morales-González

**Affiliations:** 1Laboratorio de Medicina de Conservación, Escuela Superior de Medicina, Instituto Politécnico Nacional, México, Plan de San Luis y Díaz Mirón, Col. Casco de Santo Tomás, Del. Miguel Hidalgo, Ciudad de México 11340, Mexico; nvargas_mendoza@hotmail.com (N.V.-M.); angeles_v_marcelo@hotmail.com (M.Á.-V.); eomsmx@yahoo.com.mx (E.O.M.-S.); 2Licenciatura en Nutrición, Universidad Intercontinental, Insurgentes Sur 4303, Santa Úrsula Xitla, Alcaldía Tlalpan, Ciudad de México 14420, Mexico; mtz98mauxd@gmail.com; 3Escuela Superior de Cómputo, Instituto Politécnico Nacional, Av. Juan de Dios Bátiz s/n Esquina Miguel Othón de Mendizabal, Unidad Profesional Adolfo López Mateos, Ciudad de México 07738, Mexico; jtirado@ipn.mx; 4Hospital Militar de Zona, Secretaría de la Defensa Nacional, Periférico Boulevard Manuel Ávila Camacho s/n, Delegación Miguel Hidalgo, Ciudad de México 11200, Mexico; doctorurrusquieta@gmail.com; 5Escuela Nacional de Ciencias Biológicas, Instituto Politécnico Nacional, “Unidad Profesional A. López Mateos”. Av. Wilfrido Massieu. Col., Lindavista, Ciudad de México 07738, Mexico; eduardo.madrigal@lycos.com (E.M.-B.); isela.alvarez@gmail.com (I.Á.-G.); 6Departamento de Fisiología, Laboratorio de Hormonas y Conducta, ENCB Campus Zacatenco, Instituto Politécnico Nacional, Ciudad de México 07700, Mexico; tfregoso@ipn.mx

**Keywords:** exercise training, muscle, myofibers, myocardium, silymarin

## Abstract

(1) Background: Regular exercise induces physiological and morphological changes in the organisms, but excessive training loads may induce damage and impair recovery or muscle growth. The purpose of the study was to evaluate the impact of Silymarin (SM) consumption on endurance capacity, muscle/cardiac histological changes, bodyweight, and food intake in rats subjected to 60 min of regular exercise training (RET) five days per week. (2) Methods: Male Wistar rats were subjected to an eight-week RET treadmill program and were previously administered SM and vitamin C. Bodyweight and food consumption were measured and registered. The maximal endurance capacity (MEC) test was performed at weeks one and eight. After the last training session, the animals were sacrificed, and samples of quadriceps/gastrocnemius and cardiac tissue were obtained and process for histological analyzes. (3) Results: SM consumption improved muscle recovery, inflammation, and damaged tissue, and promoted hypertrophy, vascularization, and muscle fiber shape/appearance. MEC increased after eight weeks of RET in all trained groups; moreover, the SM-treated group was enhanced more than the group with vitamin C. There were no significant changes in bodyweight and in food and nutrient consumption along the study. (5) Conclusion: SM supplementation may enhance physical performance, recovery, and muscle hypertrophy during the eight-week RET program.

## 1. Introduction

Exercise training induces a wide variety of modifications within organs and systems, and skeletal muscle and cardiac tissue engage in dynamic crosstalk regarding the mediation of adaptations during physical exercise. The two basic skeletal muscle functions comprise providing stability for body posture and contraction for body movement. The skeletal muscle mass comprises around 50–75% of the body protein pool and 40–50% of bodyweight [1]. Muscle is able to adjust to different amounts of tension when the workload increases and mechanical stress rises; then, it adapts by producing more contractile proteins along with an increase in muscle fiber size and a consequent force power [2]. Hypertrophy refers to the increase of size in pre-existing muscle fibers rather than to amplification of the cell number, termed hyperplasia; this growth comprises signaling controlled by several environmental factors, such as loading, hormones, and the availability of nutrients and energy [3]. Skeletal muscle is constituted of different types of fibers; the so-called “slow-twitch” and “fast-twitch” fibers, according to the glycolytic and ATP production, and with the ATPase activity of myosin determining the speed of muscle contraction [4]. Fibers are classified by the predominant isoforms of heavy-chain myosin. Type I corresponds to slow, and type II (IIa, IIb, and IIx), to fast [5]. Additionally, during exercise, augmented cardiac output leads to physiological and morphological heart remodeling, which also depends on the type of training (endurance/resistance), genetic factors, gender, age, and on body size and composition. Exercise induces physiologic hypertrophy and cavity enlargement in the hearts of athletes as a result of rigorous training [6]. Therefore, skeletal muscle and the heart adapt to a great volume of training.

Silymarin (SM) is a mixture of flavonoids made up of 60–80% flavonolignans; silybin is the main compound comprising around 50–70% of SM composition, in isoforms silybin A and silybin B (Figure 1). The remaining compounds include isosilybin, silydianin, silychristin, and isosilychristin, along with their respective isoforms [7]. SM is frequently known as milk thistle or St Marie’s thistle, is extracted from the seeds of *Silybum marianum*, and belongs to the *Asteraceae* family. The latter is native to the Mediterranean region but is currently distributed around the world on the majority of the continents. SM has been employed by the practice of traditional medicine since one thousand years ago in the treatment of liver disorders [8]. Dating from past years, a variety of studies have been carried out to probe the most observed properties such as antioxidant, liver protector, nephron and cardioprotective [8,9,10]. The mechanisms by which SM exerts any activity continue to be studied and some include the following: (1) free radical (FR) scavenging; (2) conducting the enzymatic and non-enzymatic antioxidant response through regulation of transcription factors such as nuclear factor erythroid 2 (NFE2)-related factor 2 (Nrf2), the coordinator for the cytoprotective and antioxidant response [11], and nuclear factor-Kappa β(NF-Κβ), which is involved on the inflammatory process; (3) inhibiting FR production by blocking the producer enzymes; (4) inducing the gene expression of protector molecules, for example, heat shock proteins (HSP), thioredoxins (Txn), and sirtuins (Sirt), conferring protection against oxidative stress [12].

The health benefits from preventing chronic diseases due to regular exercise are well recognized. In homeostatic conditions, organs and systems can mediate stress induced by physical activity [13,14]. However, when physical training exceeds the individual’s biological capacities to regulate stress, undeniable damage could affect in ubiquitous [15]. A high exercise-training load may have implications in muscle and cardiac tissues; in this manner, antioxidant supplementation in some exercise models have been tested in order to elucidate their possible effect on physical performance, recovery, or body composition improvement [16]. Nonetheless, there are no conclusive results to date. In this study, we evaluate the impact of SM consumption on bodyweight, food/nutrient consumption, and on the physical capacity of the muscle and myocardium histological modifications in a rodent model subjected to regular exercise training.

## 2. Results

### 2.1. Bodyweight

Results of changes in bodyweight are presented in Table 1. CON and ET + SM groups reported similar bodyweight gain, that is, 141 ± 9.82 g (60.72 ± 3.42%) and 142.8 ± 11.29 g (61.96 ± 4.92%), respectively. Likewise, the bodyweight obtained in the ET group was 113.4 ± 7.03 g (47.47 ± 2.93%), and that in the ET + VC group was 110.4 ± 7.49 g (47.49 ± 3.45%). Analysis among the groups indicates a significant difference between ET and ET + VC vs. CON (*p* = 0.019), but there was no difference with the ET + SM group (*p* = 0.841). In this regard, the bodyweight of the ET + SM group increased significantly when compared with those of ET (*p* = 0.042) and ET + VC (*p* = 0.046).

### 2.2. Food Consumption, Energy, and Nutrient Intake

Food consumption was registered daily before training; the previously described nutrition facts of the Chow formula 5008 label were considered for the estimation of energy, protein, and fat intake (Table 2). Average food intake per week for CON was 232.75 ± 17.7 g; the ET group reported 257.5 ± 20.69 g, ET + VC registered 239.62 ± 14.51 g, and ET + SM consumed 258.62 ± 19.95 g. When we examined the timeline, CON had the greatest food consumption in week five (332 g) and the least consumption in week eight (159 g). For the ET group, the greatest intake was reported in week three (327 g), whereas the least was in the last week (143 g). The ET + VC group registered maximal consumption in week four, while in week one, it reported minimal intakes. ET + SM had the greatest food consumption in week four (354 g) and the least in week seven; it is noteworthy that ET + SM reported the greatest food intake compared with the other groups throughout the study. Nevertheless, the statistical analysis did not reflect significant differences in the amount of food consumption among the groups over the weeks (*p* = 0.688). Energy and nutrient intake were estimated from the nutritional facts of the Formulab Chow 5008 label (PMI Nutrition International, L.L.C.) considering food consumption volume (grams consumed). Average energy, protein, and fat intake for the CON group were 387.52 ± 29.59 Kcals, 53.53 ± 4.09 g, and 15.12 ± 1.16 g; the ET group reported energy 428.73 ± 97.34 Kcals, protein 59.22 ± 4.76 g, and fat 9.29 ± 1.34 g; the ET + VC group registered energy at 398.97 ± 24.16 Kcals, protein 55.11 ± 3.38 g, and fat 13.63 ± 0.96 g, and in the ET + SM group, the energy consumed was 430 ± 33.22 Kcals, protein 57.84 ± 5.59 g, and fat 16.34 ± 1.29 g, respectively. Greatest/least energy and nutrient consumption was relative to the number of weeks of food consumption mentioned previously. The analysis revealed no significant difference in energy (*p* = 0.707), protein (*p* = 0.688), and fat (*p* = 0.698) intake among the groups during the training protocol. Bulleted lists look like this:

### 2.3. Exercise-Endurance Capacity Test

The results of the MEC test are presented in terms of time and distance in the first and last week of the training protocol (Table 3). Similar results were observed for running distance and time at week one with no differences among the groups (*p* = 0.770); however, in week eight, running time was significantly higher in the training groups compared with CON. The ET group increased 4.03 times (2.06 ± 019 min) in running time in comparison with week 1 (0.51 ± 0.09 min) and 4.12-fold with CON (0.50 ± 0.04). The ET + VC group increased 4.83 times (2.89 ± 0.13 min) in relation to week one (0.60 ± 0.05) and 5.78 times compared with CON, whereas the ET + SM group increased 5.97 times (3.7 ± 0.33 min) in comparison with week one (0.62 ± 0.07) and 7.4-fold vs. the CON group. Analysis among the groups indicated a significant difference in all trained groups and in CON (*p* < 0.001), as well as ET vs. ET + VC (*p* = 0.002) and ET + SM (*p* = 0.001). In addition, there was a significant difference between ET + VC and ET + SM (*p* = 0.045).

In relation to distance, no group exhibited significant differences during week one (*p* = 0.620). During week eight, the running distance was different in trained groups to that in the CON group. The ET group improved 5.44 times (46.11 ± 3.1 m) compared with its results in week one (8.47 ± 1.46 m) and 5.47 vs. CON (8.42 ± 1.06 m). ET + VC increased 7.12 times (71.17 ± 8.48 m) and 8.45 times in comparison with week one (9.99 ± 0.88 m) and with CON, respectively, while the ET + SM group increased 10.22 (105.23 ± 14.42 m) and 12.42 times in comparison with week one (10.29 ± 1.14 m) and CON, respectively. The analysis among the groups indicated a significant difference in all trained groups vs. CON (*p* < 0.001), as well as in ET vs. ET + VC (*p* = 0.001) and ET + SM (*p* < 0.001). There was a significant difference between ET + VC and ET + SM (*p* = 0.027) according to the statistical analysis.

### 2.4. Histological Analysis

#### 2.4.1. Quadriceps Muscle

After eight weeks of RET, histological changes can be observed in the animal quadriceps muscle of all trained groups vs. CON (Table 4). Different characteristics were considered for muscle tissue evaluation. The CON group experienced null alterations in hypertrophy, polygonal shape fiber, vascularization, splitting, and cyanophile sarcoplasm. Normal striated muscle and low inflammation levels were observed. Edo-, peri-, and epimysium were normal in size. The presence of moderate intramuscular adipose tissue can be observed (Figure 2a). On the other hand, the ET group experienced a more rounded and polygonal muscle-fiber shape, and also moderate levels of hypertrophy, a striated appearance, inflammation, and the appearance of cyanophile sarcoplasm, while vascularization reflected a low increase and splitting appeared remarkably high. There was a notable reduction of size in the endo-, epi-, and perimysium, as well as in lipid content (Figure 2b). The ET + VC group presented high hypertrophy and vascularization. Muscle fibers exhibited a more polygonal shape and a more striated appearance. On the other hand, the splitting phenomenon notably reduced inflammation, and the presence of cyanophile sarcoplasm and adipose tissue also diminished. Edo-, epi-, and perimysium were similar in size to those of the ET group (Figure 2c). Finally, the ET + SM group underwent significant changes in hypertrophy, polygonal shape, striated appearance, and vascularization. There was a noteworthy reduction of fiber splitting; simultaneously, inflammation, cyanophile sarcoplasm, and intramuscular adipose tissue presented an important diminution. The size of the endo- and perimysium became thinner, while the epimysium remained similar to CON in size (Figure 2d).

#### 2.4.2. Gastrocnemius Muscle

Evaluation of the gastrocnemius muscle after eight weeks of regular ET revealed significant changes in several histological characteristics (Table 5). The CON group did experience null hypertrophy, a polygonal fiber shape, splitting, inflammation, and cyanophile sarcoplasm. There was a moderate striated appearance in muscle fibers; medium normal size could be observed in endo- and perimysium, whereas the epimysium appeared wider. In addition, there was a low presence of peripheral satellite cells on muscle fibers; also, predominance was noticeable of red instead of white fibers (Figure 3a). The ET group underwent high hypertrophy, a polygonal fiber shape, striated appearance, and vascularization. Cyanophile sarcoplasm increased moderately, while splitting was very high. In contrast, inflammation appeared to be low. Endo- and epimysium size was smaller, whereas perimysium remained similar to CON. Intramuscular adipose tissue became reduced, but the presence of satellite cells and of white and red fibers remained similar to CON (Figure 3b). The ET + VC group experienced very high levels of hypertrophy and a highly polygonal fiber shape, striated appearance, and vascularization. Splitting was reduced to moderate; additionally, endomysium size became null, the size of epimysium was reduced to small, and that of perimysium remained the same as CON. Inflammation, intramyocellular lipid content, and cyanophile sarcoplasm reduced nearly to the baseline. Satellite cells remained null and the proportion of white fibers increased, contrary to that of red fibers (Figure 3c). On the other hand, ET + SM underwent a very high level of hypertrophy, polygonal fiber shape, and striated appearance. Vascularization became high; in addition, it was observed that inflammation, adipose tissue, and the presence of cyanophile sarcoplasm remained null, and the existence of peripheral satellite cells remained low. White fibers were moderate and red fibers were high (Figure 3d).

#### 2.4.3. Myocardium

Examination of the myocardium of the animals after the RET protocol confirms the effects of exercise and supplementation on heart architecture (Table 6). The myocardium of the CON group presented null hypertrophy and splitting; the presence of eosinophilia and vascularization was low. Endocardium size remained small, although the appearance of the pericardium was thicker (Figure 4a), while the ET, ET + VC, and ET + SM groups expressed high levels of hypertrophy and vascularization, as well as a small size of the endocardium and pericardium (Figure 4b–d). Splitting was remarkably high in the ET group, with high eosinophilia. In contrast, the ET + VC and ET + SM groups presented low-to-null myocardium splitting.

## 3. Discussion

Exercise training induces changes in the organism and a wide range of studies have shown the benefits of the regular practice of some type of physical activity in preventing or treating diseases. Exercise might lead to the adjustment of bodyweight, body composition, altered food patterns, energy, and nutrient intake. In this manner, the organism on its own possesses an amazing ability to adapt to the physical workout. Consequently, physical capacity, strength, and endurance are maximized. These latter skills are linked to histological, physiological, genetic, and metabolic alterations depending on volume, frequency, and intensity; if these are extremely high, alterations may be aggressive, giving rise to damage ion tissues and affecting athletic performance. Therefore, providing some aids to help in halting or improving damage might be a good option for physical recovery.

In this study, we attempted to examine the effect of the supplementation of SM on physical performance, muscle (quadriceps/gastrocnemius), histological changes in the myocardium, bodyweight, and food-pattern consumption after eight weeks of a physical training protocol tested in a rodent model. SM is the name given to a mix of flavonolignans obtained from the seeds of the *Silybum marianum* species [7]. SM is commonly known as St. Marie’s thistle or milk thistle due to its appearance. The uses of milk thistle have been documented for many years in the traditional medicine of several old civilizations, mainly in liver injuries [8,9,10]. A wide variety of scientific reports have approved the biological activities of SM, basically as a potent antioxidant [12]. However, the use of SM as an ergogenic in exercise-training models is limited. Choi et al. [17] determined the effect of SM on gluconeogenesis during exercise in rats and found that, after four weeks of exercise training that included running on a treadmill, the administration of SM (50 mg/kg) may improve the gluconeogenesis and β-oxidation induced by exercise training, reducing lactate and triglyceride serum levels, reducing the expression of Akt, phosphoenolpyruvate carboxykinase (PEPCK), and the peroxisome proliferator-activated receptor-γ (PPARγ) in the liver; in muscle, the expression of AMP-activated kinase (AMPK) was low and the expression of pyruvate dehydrogenase kinase 4 (PDK4) was favored. The relevance of these findings lies in that the enzyme PEPCK is involved in the first-rate limiting step of AMPK, and these are also engaged in fatty acid oxidation and gluconeogenesis and PDK4 in mitochondrial glucose oxidation. Hence, this study suggests that the administration of SM could improve glucose, lactate, and lipid metabolism in exercise.

Maximal endurance capacity can be enhanced with physical exercise, and different models in animals have been employed to prove this, including treadmill running, climbing, and swimming force, as well as others. Seo et al. [18] utilized a voluntary stand-up physical activity (SPA) model, inducing rats to lift a load equivalent to their bodyweight, while the height of the food employed in cages increased gradually for 12 weeks. After the protocol, these authors reported a reduction of bodyweight but no significant changes in evaluation of skeletal muscle mass, heart weight, food consumption, and echocardiography that compared these with the control. Nevertheless, in the MEC, the SPA group demonstrated a better time and distance in running, in addition to higher grip strength vs. the control. Histological analysis of the gastrocnemius reported minor modifications in mitochondrial content, the number of myonuclei, and muscle fibers. Many studies have attempted to demonstrate the effects of using antioxidants in sports performance, the reduction in muscle damage, the prevention of fatigue, and in the improvement of immune function [19]. These antioxidants include quercetin, vitamin C, vitamin E, resveratrol, beetroot juice, spirulina, N-acetylcysteine, and other polyphenols [19]. In fact, under normal conditions, the organism possesses its own antioxidant defenses by means of the endogenous antioxidant system; hence, it is known that a certain level of stress is necessary for the maintenance of homeostasis. The term hormesis, or exercise preconditioning (EPC), refers to how physical activity and exercise give rise to metabolic, mechanical, and oxidative stressors in skeletal muscle and the cardiovascular system. Notwithstanding this, specific doses of training provide a certain protection for future insults of injury or disease; thus, cellular systems become more efficient [20]. An increase in an exercise-induced workload confers a rise in oxidative stress as a consequence of high oxygen-cell demands, producing ROS and RNS; however, at physiological levels, these molecules modulate cell signaling and communication [21,22]. For many years, a continuous discussion has existed on whether the use of antioxidants prevents the damage to oxidative stress induced by exercise, or whether antioxidants may blunt the natural biological natural process of molecular and physiological adaptations. Therefore, these discrepancies broach the question of whether antioxidant supplementation possesses benefits or not in terms of the performance of exercise training [23]. It is noteworthy that professional athletes are often subjected to elevated energetic demands and stressors that probably surpass their cytoprotective defenses.

In fact, the International Society of Sports Nutrition at the recent document “ISSN exercise and sports nutrition review update: research and recommendations” published in 2018 [24], stated vitamin C in category I according to the classification of dietary supplements in sports, considering as follows: “I. Strong Evidence to Support Efficacy and Apparently Safe: Supplements that have a sound theoretical rationale with the majority of available research in relevant populations using appropriate dosing regimens demonstrating both its efficacy and safety”. Besides, the International Olympic Committee on its “IOC consensus statement: dietary supplements and the high-performance athlete” [25] refers to vitamin C as a nutrient that may scavenge ROS to reduce oxidative stress and enhance immunity, and reduce cortisol and interleukin-6 responses in exercise of humans. Moreover, IOC considers vitamin C as part of the nutrients that could support with recovery, injury management, muscle soreness, and training capacity.

Nevertheless, previous studies have shown the effect of vitamin C supplementation on physical performance, oxidative stress, and the antioxidant status of athletes, but the results are controversial. The majority of these point out that supplementation may interfere in cell-signaling responses for exercise adaptation [26]. Teixeira et al. [27] found that four weeks of supplementation with a mixture of antioxidants (α-tocopherol 272 mg, β-carotene 30 mg, lutein 2 mg, selenium 400 μg, zinc 30 mg, and magnesium 600 mg) including vitamin C (400 mg) in athletes did not provide protection against inflammation and lipid peroxidation. Even more so, it may interrupt muscle recovery when compared with that of the placebo group. The results were divided, in another study after a blood screening with regard to low plasma vitamin-C levels (35 ± 8 μmol/L) and high-plasma vitamin-C levels (78 ± 8 μmol/L) after one month of supplementation with 1000 mg/day, benefits in terms of physical performance were observed only in the low-plasma vitamin-C group, increasing the maximal oxygen uptake (VO_2max_) by 14%. At the same time, oxidative stress biomarkers were reduced (urine F-isoprostanes by 18%, and plasma protein carbonyls by 23%). The high-plasma vitamin-C group did not exhibit significant changes [28]. Thus, it would be reasonable to assume that there is a benefit in vitamin-C supplementation only within the context of a deficiency. On the other hand, the investigation with senescence marker protein 30 (SMP-30) knockout mice mimicking a defect in the biosynthesis of L-Ascorbic Acid (AA) divided into the AA-sufficient group (AA(+)) received 1.5 g/L of AA and AA-deficient (AA(−)) group received tap water (free AA), revealing that, after four weeks of treatment, AA content or gastrocnemius muscle was considerably lower in the AA(−) group (0.7%) compared with the AA(+) group. In addition to that, physical performance assessed by a treadmill test was significantly less in the AA(−) group during weeks 4, 12, and 16 (20, 50, and 46%, respectively) in contrast to that of the AA(+) group. Additionally, the cross-section area (CSA) of the soleus muscle of mice was significantly lower at 16 weeks, as well as the weight of the muscles observed at 12 and 16 weeks in the AA(−) group. The investigation suggests that AA deficiency is associated with muscle wasting, but this may return to normal with the reposition of AA [29].

Concerning the MEC results, we found that regular exercise induces physiological adaptations and leads to the improvement of the capacity of executing a physical workout with a control of time, frequency, and intensity. The groups with antioxidant, SM, and vitamin C supplementation had better results in terms of running time and distance. Furthermore, the ET+SM group exhibited a better physical performance: comparison among the groups did show a significant difference between ET + SM and ET + VC, due to SM increasing running time 1.3-fold times above that of vitamin C. Duan et al. [30] tested the pre-workout oral ingestion of luteolin-6-C-nehoesperidoside (LN) (25, 50, and 75 mg/kg, and ascorbic acid (AA) (100 mg/kg) to evaluate the anti-fatigue effect on a forced swimming test (FST). The study was performed in old male rats (20–22 weeks of age). After three weeks of FST, dose-dependent LN enhanced the swimming ability, increasing the exhausting swimming time in LN75, LN50, and LN (28.6 ± 2.1, 24.8 ± 1.9, and 21.6 ± 1.2 min, respectively) as compared with the trained group without supplementation (12.9 ± 12 min). Histological analyses of the muscle and liver cellular morphology of all trained groups revealed alterations such as inflammation, edema, and congestion. A swelling disorder in muscle and liver tissues could also be detected, plus an increased number of foam cells. A notable diminution of these pathological issues was observed in all LN-administered groups in a dose-dependent manner with no differences in the AA group. Askari et al. [31] suggested that supplementation with quercetin (500 mg/day) and vitamin C (200 mg/day) for eight weeks may aid in reducing muscle-damage markers and the body-fat proportion, but they did not appear to affect physical performance in a treadmill time-to-exhaustion test. Similar research by Causo et al. [32] demonstrated that 25 mg/kg of quercetin did not affect endurance time-to-exhaustion, nor the VO_2max_ peak after six weeks of endurance training and quercetin supplementation in a rodent model. However, after high-intensity training, quercetin increased blood-lactate production, probably caused by enhanced glycolysis.

Exercise training gives rise to alterations in nearly the entire organism as a result of stress-induced adaptation, but muscles are the tissues exhibiting the most notorious modifications. Muscle is an extremely plastic tissue that is able to adapt to contractile activity. Endocrine variations, cell lineage, and metabolic, physiological, and nutritional factors can exert an influence on muscle features and plasticity [33]. Exercise induces mechanical loading, triggering different responses such as hypertrophy, which is characterized by the increase of fiber size rather than the number of myofibers or cells (hyperplasia) [34]. Hypertrophy involves protein synthesis, avoiding degradation in multiple anabolic processes. Myofiber protein synthesis is controlled via DNA codification in myonuclei; each myonucleus engages in mRNA transcription and protein synthesis in sarcoplasm, occupying a specific volume. Then, when protein synthesis is elevated, the cell volume and CSA of muscle fibers grow, expanding the nuclear domain beyond normal boundaries. This process requires the fusion of new myonuclei to myofibers [35]; thus, there is a proportional relationship in the amount of myofiber and in the number of myonuclei [36]. In the present study, histological changes were notable in all training groups, but more so in those with antioxidant supplementation. The quadriceps and gastrocnemius muscles of animals subjected to RET underwent an important modification of muscle-fiber architecture; hypertrophy, polygonal shape, and striated appearance were relevant, as well as vascularization. However, the histological changes are remarkable in ET + SM as compared with the CON and ET groups. Some evidence suggests that antioxidants may exert an impact on muscle recovery and vascular function with exercise training. On the other hand, current data is not completely clear with regard to the ergogenic effect on exercise training [37].

A reduction of size was observed in the endo-, epi-, and perimysium in the quadriceps muscle in all trained groups (Figure 2). However, the size in ET + SM and ET + VC remained similar to that of the ET group. In gastrocnemius muscles, it was observed that endo- and epimysium size was slightly compacted in the ET group. On the other hand, there was a significant reduction of size in the ET + VC and ET + SM groups compared with the CON group, but there was no difference between SM and vitamin-C treated groups. On the other hand, the perimysium in gastrocnemius muscle did not change in any group. Muscle fibers are packed in peri- and endomysium fascicle bundles and the whole muscle is bound by epimysium fascia formed basically by connective tissue. This connective tissue can also contribute to muscle size and strength because it is able to adapt and change according to the mechanical workload [38]. We were able to observe that the connective tissue became thinner as a result of the increase in muscle-fiber size and polygonal shape. It could be observed that hypertrophy was the result of an increased workload in response to the demands of the exercise.

In addition, the histological results of the quadriceps muscle must be highlighted with respect to the splitting phenomenon, together with inflammation and the presence of cyanophile sarcoplasm in the ET group. Simultaneously, gastrocnemius splitting becomes very prominent and the appearance of cyanophile sarcoplasm is somewhat less visible compared to CON. Contrary to the quadriceps muscle, in the gastrocnemius the phenomenon of inflammation remained moderate. Splitting is defined as the approximate proportioned division that may or may not run the entire extent of the fiber and that occurs at a level of the myofibril [39]. This phenomenon is normally due to excessive overload, together with the branching characterized by the emergence of smaller fibers stemming from a larger one. Both splitting and branching may contribute to muscle recovery, forming new fibers and increasing CSA as a result of extreme overload. Some studies reported an increase in the number of muscle fibers resulting from branching or splitting after stretch overload [40,41]. Additionally, some of this evidence supports the idea that prior to satellite cells being stimulated to migrate into extra-fascicular space from the basal lamina, an apparent degeneration is considered necessary for the creation of nescient fibers [42,43]. In resistance-strength sports such as bodybuilding or powerlifting, histochemical and immunohistochemical analyses of muscle biopsy samples have reported fiber splitting, importantly resulting in hypertrophy [44,45]. Repeated exposure to muscle-eccentric actions in high-performance elite athletes who are often subjected to extreme and prolonged loading leads to fiber splitting, regeneration, myocyte grafting, and pronounced hypertrophy [39]. It is important to consider that hypertrophy is basically an anabolic process: therefore, the appropriate environmental conditions of oxygen, nutrients, hormones, sleeping/resting time, and certain other elements are required to achieve muscle growth. Unless these aspects are assembled, excessive loading might blunt hypertrophy [46]. In this research, we could observe that fiber splitting was significant in the quadriceps and gastrocnemius muscles of the ET group and led to moderate hypertrophy; on the other hand, reports on ET + SM cited low splitting but high hypertrophy. It can be assumed that SM could improve environmental conditions for muscle growth, probably due to the reduction of stressors such as free radicals in the cellular environment.

The inflammatory response to exercise (IRE) is commonly observed in strenuous exercise with imminent signs of tissue damage, cellular infiltrates, acute-phase response, leukocyte activation, the release of inflammatory mediators, complementary system activation, free radical production, and fibrinolytic and coagulation cascades [47]. Histological analyses provide information on leukocyte activation, with macrophages and TCD4 lymphocytes, some B lymphocytes, and activated polymorphonuclear cells in terms of the damage in a muscle biopsy induced as a result of exercise [48]. The producers of leukocyte activation, such as interleukin-1β, prostaglandins, proteases, and oxidants, may take part in tissue injury [49]. Previous findings support the theory that mononuclear cell infiltrates participate in muscle repair to a greater extent than in the development of tissue injury [50]. Further, pro-inflammatory mediators such as cytokines, interleukin-6 (IL-6), and tumor necrosis factor-α (TNF-α) are released in response to exercise depending on the volume and intensity of the training bout. The study of SM as ergogenic in the performance of exercise or in muscle recovery is, to our knowledge, null. Some findings indicate that SM and its major compound, silibinin, could act as a protector in ischemic reperfusion injuries in tissue types including skeletal muscle [51] through a different mechanism, such as enhancing the antioxidant response, scavenging free radicals, inhibiting inflammatory cytokines, and cell death, among others [52]. Barari et al. [53] evaluated the effect of SM alcoholic juice (5 mg/kg/day) in 24 males divided into three groups: exercise without SM(E); exercise plus SM (ESM), and control (C). These individuals were subjected to an endurance training program of 75 min at 70% of heart-rate reserve three times a week, and afterward, two weeks of orally ingested SM juice and training. A reduction was found in the number of neutrophils in the E and ESM groups, while lymphocytes significantly increased in the ESM group. There was a significant increase in IL-6 in both groups, while TNF-α did not change in the E group. IL-6 was related to low growth hormone (GH) and insulin-like growth factor1 (IGF-1), which are involved in muscle growth. This suggests that SM juice does not differ in terms of the reduction of inflammation and that it perhaps interferes in muscle growth. It is possible that the dose used in this research would be insufficient to produce any positive effect on inflammatory and anabolic biomarkers. In comparison with the results of our research, visible change in fiber size and hypertrophy can be observed, as well as the diminution of infiltrating cells in the muscle tissue of animals with SM consumption.

As Figure 2c,d and Figure 3c,d indicate, the quadriceps and gastrocnemius muscles of the ET + SM group had a more rounded polygonal fiber shape, especially in terms of the ET + VC group; also, the fibers appeared to be more hypertrophied and striated. In the quadriceps muscle of the ET + SM group, splitting, inflammation, and cyanophile sarcoplasm were as remarkably low as CON. In the gastrocnemius, the splitting phenomenon was dramatically reduced compared with the ET group vs. the ET + VC group; moreover, inflammation with cyanophile sarcoplasm decreased as in the CON group and in the SM-treated group. The relation has been previously reported in microscopy analyses between chronic-resistance exercise loading and splitting along the length of the muscle fibers, representing a possible increase in fiber quantity and a decrease in mean CSA [54]. In addition, vascularization in the quadriceps muscle was higher in the SM-treated group vs. that of the other groups; on the other hand, in the gastrocnemius in ET + VC and ET + SM, it was comparable to the ET group but significantly higher than in CON.

Likewise, there was a reduction of intramuscular fat storage in both the quadriceps and gastrocnemius muscles in ET compared with CON. In the case of animals supplemented with vitamin C and SM, modifications in lipid content were significantly lower. Otherwise, the decrease of intramuscular fat storage was comparable between the ET + VC and ET + SM groups. This could probably be due to a better utilization of lipids in active muscles, inducing changes in bodyweight and composition, which will be discussed further.

In the present work, we detected the presence of satellite cells in the gastrocnemius, but not in the quadriceps. The appearance of these cells was low, as in CON. Satellite cells are stem cell populations in muscles: the activation and proliferation of these cells is associated with hypertrophy. These are located on the outer surface of myofibers, between the basal lamina and the sarcolemma, which is why they are called satellite cells. Exercise-induced mechanical load gives rise to activation, proliferation, and fusion in existing fibers in order to donate nuclei in situations of damage and protein synthesis, as reported previously [55]. Consequently, there is an increase in contractile protein (actin and myosin) and myofiber size, but not the number of muscle fibers [56]. The histological study of myofibers indicates that hypertrophy takes at least 8–16 weeks of training in normal muscles. Nonetheless, previous studies suggest that hypertrophy is induced more efficiently and faster in atrophied than in normal muscle. Itoh et al. [57] determined, with an atrophy model of tail suspension (TS) in mice, the effect of stand-up exercise (SE) as a model of RT in the recovery of atrophied muscles. After seven days of SE training, the CSA soleus muscle was nearly recovered like the control group. The results proved the increase of myonuclei at 1.5- and 1.6-fold vs. control and non-trained groups, respectively, after four days of SE. In addition, the proliferation of satellite cells began two days after the initiation of training, and fusion with pre-existing myofibers, two days later. This suggests that the proliferation and activation of myogenic satellite cells occurs within a period of two to four days after training stimuli on atrophied muscles. It has been observed that stimuli by RT initiate myoblast proliferation and blending with existing fibers. Subsequently, the proliferation of satellite cells raises the myonuclei content.

Later, these same authors [58] studied the relationship between resistance training (RT) and the recovery of atrophied muscle linked to the number of myonuclei. These authors found that 14 days of tail suspension are sufficient to induce atrophy in mice, and they then investigated different intensities of maximal isometric contraction (MC) (10, 40, 60, and 90% of MC) for the period of seven days of RT. The results showed that at 40 and 60% of MC and the CSA of muscle fibers were facilitated. Contrariwise, extremely low or high RT (10 or 90% of MC) does not promote recovery and maximal isometric contraction. Fibers with a lower size were found at 60 and 90% of MC; conversely, fibers with larger sizes were identified at 40 and 60% of MC. Additionally, at 40% of MC, there was an increase in the number of myonuclei and the activation of myogenic satellite cells linked to the recovery of muscle fibers. On the other hand, exercise at 90% of MC also promotes the activation of satellite cells, but damaged environmental conditions do not allow the cells to survive. As a result, the number of myonuclei remains low; therefore, the number of muscle fibers in CSA did not increase. In conclusion, the fusion of myogenic satellite cells with RT-stimulated neo-fibers at a moderate load induced the formation of myonuclei, expanding the CSA and promoting recovery from atrophy. We could detect that the presence of visible satellite cells remained as in CON in the gastrocnemius and was not visible in the quadriceps. It is probable that more specific techniques for satellite-cell identification are needed.

Regarding fiber types, in this study we were able to observe that gastrocnemius-muscle analysis in the ET + VC group expressed more white than red fibers, contrary to ET + SM, which expressed more red instead of white fibers. Skeletal muscle is composed of different types of fibers that are classified based on the speed of contraction as fast- or slow-twitch fibers. Myosin heavy chain (MyHC) isoforms are a key factor in determining the velocity of shortening; taken together, fibers are categorized as type I, type IIx/d, and type IIa, considering that myosin ATPase activity determines the speed of interaction between actin and myosin. Thus, slow-twitch fibers are related to myosin ATPase type I and fast-twitch fibers, to myosin ATPase type II [5]. Type I fibers have a greater proportion of myogenic satellite cells compared to type II fibers. Consequently, red muscles have a greater potential to form new fibers than white muscles. In addition, the distribution of the muscle-fiber-type population determines muscle-fiber composition. For instance, the result of ATPase activity demonstrated that the gastrocnemius contains around 50% of slow-twitch fibers and the vastus lateralis, about 32%. On the other hand, the soleus and vastus intermedius muscles contain a 70 and 47% proportion of slow-twitch fibers [59]. Transcription factors such as STAT5a/b have played a critical role in the growth and maintenance of the muscle mass function. The study in mice with a specific deletion in skeletal muscle of *Stat5a/b* genes (Stat5MKO) demonstrated a significant modification in gene expression that involved fiber type and muscle growth in quadriceps. The lack of STATa/b renders possible the amplified expression of several genes related to the synthesis in novo of type I fibers, considering that muscle is composed mainly of type II fibers [60]. On the other hand, the nature of training also determines the muscle-fiber composition. It is known that endurance exercise involves relatively more slow-twitch than fast-twitch fibers; in contrast, athletes participating in resistance training, such as sprinters, have predominantly more fast-twitch fibers building their muscles [61,62,63]. The type of training protocol employed in this investigation was predominantly endurance exercise; however, the different degrees of inclination on the treadmill utilized during training drove the animals to engage in an effort comparable to uphill running, which is different from running on flat ground; therefore, the muscle work implied the parallel activation of the majority of the fiber types involved in the mechanical load. It continues to remain unclear why vitamin C and SM stimulate a dissimilar differentiation of fiber types.

Long-term exercise training coordinates a variety of adaptations in the cardiovascular system and encourages heart remodeling in response to chronic workload, promoting growth in the absence of cardiomyocyte proliferation. Oxygen demands rise considerably during exercise, especially during intense bouts; as a consequence, there is a remodeling of the heart. The hypertrophy induced is a physiological growth known as “athlete’s heart”, which is different from pathologic hypertrophy [64]. Exercise stimulates the blood pressure and flow redistribution up to 10-fold, which implicates the release of active molecules for the dilation of vessels, ensuring that oxygen and nutrient-rich blood will reach active tissues, basically skeletal muscle [65]. Heart remodeling also depends on the type of exercise-training endurance or strength required to meet the particular demands of each of these. Endurance training requires continued heart blood output, which implies a dynamic active cardiac function, whereas static exercise, such as strength- or resistance-training, produces short loads of shunting blood from the heart to active muscles. Other sports comprise a combination of dynamic and static activities, leading to both types of hemodynamic adaptations in the cardiovascular system. In this study, the myocardium entertained interesting histological variations, as in the previously described muscles. Compared to the CON group, the ET, ET + VC, and ET + SM groups reported the same level of hypertrophy and vascularization. Preceding reports noted that three to four weeks of treadmill training are sufficient to induce hypertrophy of the myocardium and total cardiac mass as a result of physiological growth. It is the increase in the size of a cardiac-cell group, and not the cardiac-cell quantity, that produces real heart-adaptative remodeling [54]. In addition, in physiological cardiac growth, the vascular system undergoes physiological and morphological changes; the increased heart volume ejected from the heart triggers molecular signaling pathways for the creation of new vessels [66]. As mentioned previously, the training type elicits the adaptation to the hemodynamic load; for instance, endurance exercise entails a constant supply of oxygen-rich blood, promoting angiogenesis and “internal capillarization”, in reference to the formation of new blood capillaries within the muscle fibers, the greater capillary supply supporting hypertrophy [67]. In fact, skeletal-muscle capillarization, together with the activation of satellite cells, is a determinant for the renewal of muscle mass in adults [68]. Internal capillarization is more frequent in type I muscle fibers due to the predominant oxidative metabolic demands [5]. The treadmill-training protocol employed is mostly a type of endurance training; hence, the physiological and morphological adaptations characteristic of this type of exercise were distinguished. The authors were able to observe close coordinate changes in quadriceps/gastrocnemius muscles and in the heart in all trained groups in this experiment. It appears that SM consumption promoted the cardiac adaptative remodeling observed as hypertrophy and vascular development in skeletal muscle and myocardium. Moreover, it is reasonable to assume that SM consumption enhances MEC in the promotion of cardiac function along with capillarization within the fibers and optimal oxygen-nutrient blood delivery to support muscle contraction with less damage when compared to the ET group alone.

Simultaneously, the ET group exhibited a considerable increase of fiber splitting and eosinophilia, but ET + VC and ET + SM revealed the decline of these. It is possible that this antioxidant may reduce inflammatory responses in heart tissue due to RET. In this regard, the effect of SM on physiological changes in the myocardium in exercise-training protocols has not, to our knowledge, been explored to date. Some studies provided evidence of the protective effects of silibinin, the major component of SM, in different tissues affected by ischemia-reperfusion injuries (IRI). The main mechanisms described for the protective effects of SM against IRI include decreasing inflammatory cytokines, inducing the expression of the endogenous antioxidant system, inhibiting cell death, and scavenging free radicals [69,70]. SM has a long story related to the treatment of liver disorders, but at present, more attention is being dedicated to investigate its possible benefits in other organs, such as the heart. Wide-ranging models of ischemic heart damage have been conducted in pre- and post-conditioning testing in SM effectiveness on preventing or treating damage that occurs upon reperfusion, concluding that SM and its components appear to influence signaling pathways involved in pre-conditioning and inflammatory activity and may possibly exert an impact on protecting heart tissue from IRI [71]. However, such effects have been minimally elucidated in exercise-training protocols.

Other modifications observed in cardiac tissue included that endo- and pericardium diameter was reduced when compared to the CON group. The endo- and pericardium were similar in ET, ET + VC, and ET + SM groups. It is assumed that SM and vitamin C did not significantly alter the surrounding connective tissue in comparison to ET alone. 

Regarding body mass, it was observed that, after eight weeks of RET, the total bodyweight gain of the CON and ET + SM groups was significantly higher than that of the ET and ET + VC groups. ET + SM reported the maximal increase, contrariwise, ET and ET + VC similarly reported the lowest increase. Total bodyweight gain in the CON and ET + SM groups was equally proportional; however, histological analysis revealed the aforementioned hypertrophy and an important reduction of intramyofibrillar lipid storage when ET + SM was compared with the CON group. Despite that very few studies have already been conducted to probe the ergogenic effect of SM on exercise, some evidence showed changes in bodyweight with SM treatment under certain conditions. Alozy et al. [72] detected a significant increase in bodyweight gain (*p* < 0.05) in rats treated with 200 mg/kg bodyweight alone and in combination with 5 mg/kg/day of cyclosporine after 15 days of treatment and continued for periods of 30 and 45 days. The group exclusively treated with SM re-registered the highest bodyweight gain. Together with the reduction of the biochemical parameters of blood glucose, cholesterol, alanine aminotransferase (ALT), aspartate aminotransferase (AST), malondialdehyde (MDA), phosphatase alkaline (PHA), and the decrease in liver-histopathology indicators, suggest the hepatic protection of SM against cyclosporine-induced damage. Thus, this can be conceived of as a correlation between bodyweight gain and the increase of muscle mass in the group treated with SM. At the same time, food consumption was very similar in all groups; ET + SM registered the highest food intake, despite there being no significant differences in the pattern of food ingestion. In addition, energy and nutrient intakes were proportional to food ingestion. It is striking that the composition of the animals’ diet was high in protein (23%), which is a key factor in inducing muscle growth. It is well recognized that a positive nitrogen balance deriving from a protein source with a good proportion of essential amino acids, particularly leucine, creates, together with other elements, an anabolic environment for growing muscle mass [73]. Additionally, SM has come to be recognized as a protein synthesis stimulator, because it is able to enter the nucleus and enhance DNA synthesis, along with the transcription of rRNA, by stimulating RNA polymerase I enzymes. Therefore, the synthesis of structural and functional protein is improved. The availability of more transporters and enzymes also promotes cell regeneration [74]. In this case, SM administration may improve cellular conditions, thus possibly causing the highlighted modifications in body composition and weight gain. However, the results of this investigation must be accompanied by other biochemical and molecular markers to support this theory.

## 4. Materials and Methods

### 4.1. Animals and Experimental Design

Eight-week-old male Wistar rats with a bodyweight of around 200–220 g were adapted in environmentally control facilities with a 12-h:12-h light/dark cycle and a temperature of 22 °C for 1 week prior to the experiment. Rats were fed a high energy and protein diet (23% protein, 6.5% fat, 4% fiber, 12% humidity, and 8% ash) Chow formula 5008 (PMI Nutrition International, L.L.C.) and water ad libitum. All procedures were approved by the Comité Interno de Bioética of the Instituto de Ciencias de la Salud, Universidad Autónoma del Estado de Hidalgo, Mexico, with approval number CIECUAL/012/2019. All procedures were performed according to the Official Mexican Guidelines for Laboratory Animal Use and Care (NOM-062-ZOO-1999).

Animals were randomly divided into four groups (*n* = 5) as follows: sedentary control group (CON); exercise training group without supplementation (ET); exercise training and vitamin C group (ET + VC); exercise training plus silymarin group (ET + SM). ET + VC and ET + SM groups received 100 mg/kg of vitamin C and silymarin 100 mg/kg, 30–40 min prior to the training session. Vitamin C and SM were administered intragastrically (i.g.). Vitamin C has been consider as control because of the antioxidant activity, recovery and immune activity reported in several studies of exercise training models in both animal and humans according with references [24,25].

### 4.2. Exercise-Training Model

The regular exercise training (RET) design is based on mimicking high-performance training based on athletes in a previous training protocol [75]. Animals were subjected to an adaptation of the physical-training phase of the treadmill (Motor Treadmill Pro-form 305 CST adapted for rodent) as follows: day one, animals ran 15 min at a speed of 16.6 m/min 0 degrees of inclination; day two, animals ran 15 min at a speed of 16.6 m/min and 0 degrees of inclination followed by 15 min at 18.33 m/min and a 5-degree inclination; day three, animals ran 10 min at a speed of 16.6 m/min and an inclination of 0 degrees, followed by 10 min at 18.33 m/min and at 5 degrees of inclination, and ending with 10 min more at 20 min/min and at 10 degrees of inclination. After the adaptation phase, rats trained five days per week with the following protocol: 10 min at 16.6 m/min and 0 degrees of inclination followed by 10 min at 18.33 m/min at 5 degrees of inclination and 20 min/min at 10 degrees of inclination until completing a 60-min training session.

### 4.3. Food Intake and Bodyweight Control

Bodyweight was registered at the beginning and at the end of the experiment and was monitored weekly prior to the exercise training session. Food intake was measured daily before the training session and the amount was registered.

### 4.4. Exercise Endurance-Capacity Test

Maximal endurance capacity (MEC) was determined during week one of the training protocol and during the last week of the protocol. The test consisted of incremental treadmill running, was started with 16.66 m/min, and the speed was gradually increased at 1.67 m/min every 30 s at 0 dof inclination until exhaustion [18]. The MEC was evaluated in terms of running distance and time.

### 4.5. Sample Collection

After the last training session in week eight, animals were sacrificed by cervical dislocation prior to being anesthetized with phenobarbital sodium (40 mg/kg). The quadriceps, gastrocnemius muscles, and heart tissue were obtained and embedded in formaldehyde solution (10%) to preserve them, and they were stained with Hematoxylin and Eosin (H&E); this latter technique will be explained further ahead.

### 4.6. Histological and Microscopy Analyses

Samples for histological analyses were processed for H&E staining according to the specifications, which are briefly described [76]. Samples were fixed, cut, and paraffin-embedded. Glass slides holding paraffin sections were placed in staining racks. Samples were cleaned of paraffin in three changes of xylene for 2 min per change. After this, samples were hydrated, slides were transferred through three changes of 100% ethanol for 2 min per change, and then samples were transferred into 95% ethanol for 2 min and 70% in ethanol for 2 additional min. Slides were washed in running tap water for 2 min at room temperature. The samples were stained with a Hematoxylin solution for 3 min and again placed under running tap water for 5 min at room temperature. Afterward, the slides were stained with eosin Y solution for 2 min and then subjected to dehydration by dipping the slides in 95% ethanol 20 times, after which the slides were transferred into 95% ethanol for 2 min and additionally transferred once again throughout two changes in 100% ethanol for 2 min each change. Finally, samples were cleared in three changes of xylene for 2 min per change, and a drop of Permount was placed above the tissue on each slide with a coverslip placed on it. Samples were viewed with an electronic microscope (Olympus CH30). Tissue analyses were performed based on the criteria of Heffner et al. [77] for muscle and on those of Goldblum [78] for myocardium. Each tissue was analyzed for specific characteristics. Quadriceps muscle included (1) hypertrophy, (2) polygonal fiber shape, (3) striated appearance, (4) endomysium, (5) epimysium, (6) perimysium, (7) vascularization, (8) splitting, (9) inflammation, (10) adipose tissue, and (11) cyanophile sarcoplasm. Gastrocnemius muscle included (1) hypertrophy, (2) polygonal fiber shape, (3) striated appearance, (4) endomysium, (5) epimysium, (6) perimysium, (7) vascularization, (8) splitting, (9) inflammation, (10) adipose tissue, (11) cyanophile sarcoplasm, (12) satellite cells, (13) white muscle fibers, and (14) red muscles fibers. Myocardium included (1) hypertrophy, (2) endocardium, (3) pericardium, (4) vascularization, (5) splitting, and (6) eosinophilia.

### 4.7. Statistical Analysis

All data are expressed as mean ± standard error (SE). Changes between groups were evaluated utilizing a Student *t*-test and a one-way analysis of variance (ANOVA). Data analysis was performed with SigmaPlot ver. 14.0 statistical software. Differences were considered statistically significant at *p* < 0.05.

## 5. Conclusions

The organism possesses the ability to mediate stressors when exercise is engaged in, yet a chronic high volume of training may surpass the capacity of the organism’s endogenous modulation, consequently inducing damage at different levels. Damage can be explained as more frequent lesions, extra time for recovery, less physical capacity, and alteration of the immune system response. In this investigation, the impact of SM was evaluated in terms of physical capacity, histological changes in quadriceps, gastrocnemius, and myocardium, bodyweight modification, and food, as well as an energy pattern of consumption, during a RET program in a rodent model. Notwithstanding this, there is no consistent current data on antioxidants use with respect to sports performance. We found that SM consumption improves damage to skeletal muscle and cardiac tissue by reducing fiber splitting and inflammation. In addition, hypertrophy was enhanced, probably due to the superior vascularization that allowed for proper oxygen and nutrient delivery to muscle fibers, although bodyweight and food and energy consumption did not change significantly during the training protocol. Endurance capacity improved in all trained groups, and it was clear that exercise brought about physiological adaptations after eight weeks of regular training, enhancing energy and the oxygen supply in the rodents’ organisms. SM improved running time and distance more than in the other groups.

In conclusion, it can be assumed that SM consumption benefitted physical performance, induced quadriceps/gastrocnemius muscle and myocardium recovery and remodeling and, furthermore, body composition, by growing lean muscle mass for the eight-week period of the exercise-training program. On the other hand, we probably cannot arrive at definitive criteria on the use of SM as ergogenic in sports because further studies are needed to explain the possible molecular mechanism by which SM interacts, inducing the observed effects. Finally, there is currently an open door for future investigation in basic science regarding the study of ergogenic substances as supplements that can be potentially used for physically active individuals to achieve a better physical-athletic performance safely and with the least possible risk.

## Figures and Tables

**Figure 1 ijms-21-07724-f001:**
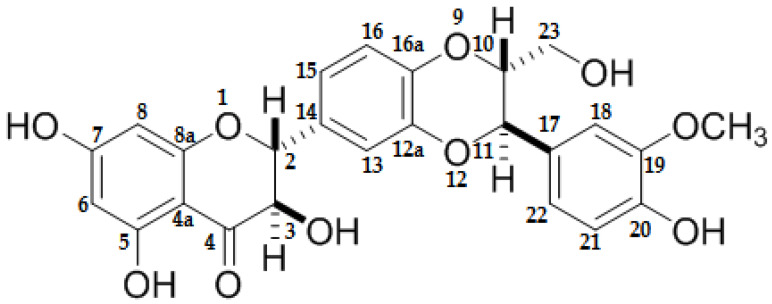
Silybin structure the major compound of silymarin.

**Figure 2 ijms-21-07724-f002:**
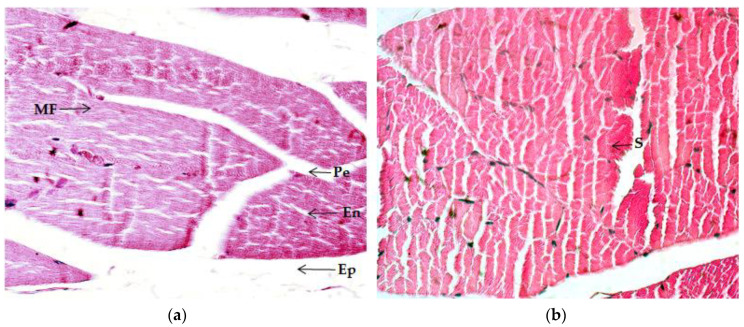
Quadriceps muscle histology (Hematoxylin and Eosin (H&E) stain (40×). (**a**) Image of the control group. (**b**) Image corresponding to the exercise-training groups without supplementation. (**c**) Image of the exercise-training groups plus vitamin C. (**d**) Image of the exercise-training group plus silymarin. MF: muscle fiber; En: endomysium; Ep: epimysium; Pe: perimysium; S: splitting; I: inflammation.

**Figure 3 ijms-21-07724-f003:**
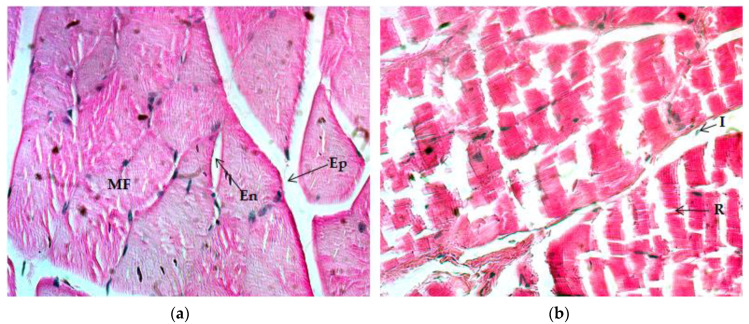
Histology of the gastrocnemius muscle, Hematoxylin and Eosin (H&E) stain (40×). (**a**) Image of the control group. (**b**) Image of the exercise-training group without supplementation. (**c**) Image of the exercise-training groups plus vitamin C. (**d**) Image of the exercise-training group plus silymarin. MF: muscle fiber; En: endomysium; Ep: epimysium; S: splitting; I: inflammation.

**Figure 4 ijms-21-07724-f004:**
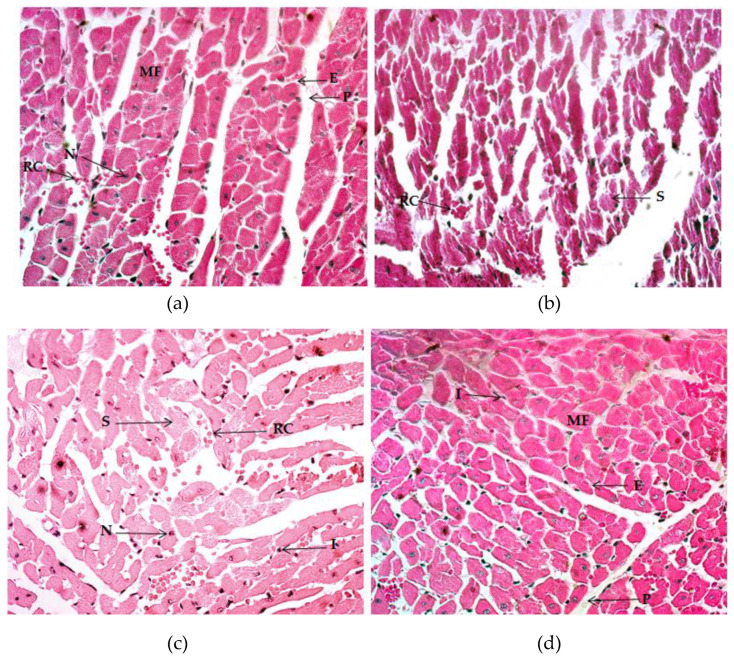
Histology of the myocardium, Hematoxylin and Eosin (H&E) stain (40×). (**a**) Image of the control group. (**b**) Image of the exercise-training group without supplementation. (**c**) Image of the exercise-training groups plus vitamin C. (**d**) Image of the exercise-training group plus silymarin. MF: muscle fiber; E: endocardium; P: pericardium; RC: red cells; N: nuclei; S: splitting; I: inflammation.

**Table 1 ijms-21-07724-t001:** Comparisons in bodyweight gain comparisons expressed in grams and percentages.

Group	Initial Weight (g)	Final Weight (g)	Bodyweight Gain (g)	Percentage of Weight Gain (%)
CON	231.8 ± 7.06	372.8 ± 15.22	141 ±9.82	60.72 ± 3.42
ET	239.0 ± 2.76	352.4 ± 7.6	113.4 ± 7.03 ^a,c^	47.47 ± 2.93 ^a,c^
ET + VC	233.4 ± 8.37	233.4 ± 11.71	110.4 ± 7.49 ^a,c^	47.49 ± 3.45 ^a,c^
ET + SM	230.8 ± 4.37	373.6 ± 12.07	142.8 ± 11.29 ^b^	61.96 ± 4.92 ^b^

Values are expressed as the mean ± SE in each experimental group (*n* = 5). ^a^
*p* < 0.05 vs. control group (CON); ^b^
*p* < 0.05 vs. exercise training group(ET); ^c^
*p* < 0.05 vs. exercise training plus silymarin group (ET + SM).

**Table 2 ijms-21-07724-t002:** Report of average food intake, energy, protein, and fat consumed per week.

Group	Food Intake (g)	Energy (Kcals)	Protein (g)	Fat (g)
CON	232.75 ± 17.77	387.53 ± 29.59	53.53 ± 4.09	15.12 ± 1.16
ET	257.5 ± 20.69	428.73 ± 34.46	59.22 ± 4.76	16.73 ± 1.34
ET + VC	239.62 ± 14.51	398.97 ± 24.16	55.11 ± 3.38	15.63 ± 0.96
ET + SM	258.62 ±19.95	430.61 ± 33.22	59.48 ± 4.59	16.81 ± 1.29

Normal distribution values are expressed by the mean ± SE. ANOVA for analysis between the groups reported no significant differences in food intake, energy, protein, and fat (*p* > 0.05) over the eight weeks or regular exercise training.

**Table 3 ijms-21-07724-t003:** Results of the exercise-endurance capacity test (time and distance) in the first and last weeks of the training program.

Week	CON Time (min)	ET	ET + VC	ET + SM
**1**	0.58 ± 0.06	0.51 ± 0.09	0.60 ± 0.05	0.62 ± 0.07
**8**	0.50 ± 0.04	2.06 ± 0.19 ^a^	2.89 ± 0.13 ^a^^,^^b^^,^^c^	3.70 ± 0.33 ^a^^,^^b^
	**Distance (m)**			
**1**	9.70 ± 0.98	8.47 ± 1.46	9.99 ± 0.88	10.29 ± 1.14
**8**	8.42 ± 1.06	46.11 ± 3.10 ^a^	71.17 ± 8.48 ^a^^,^^b^^,^^c^	105.23 ± 14.42 ^a^^,^^b^

Values are expressed as the mean ± SE in each experimental group (*n* = 5). ^a^
*p* < 0.001 vs. the control group (CON); ^b^
*p* < 0.05 vs. exercise training group (ET); ^c^
*p* < 0.05 vs. exercise training plus silymarin (ET + SM) group.

**Table 4 ijms-21-07724-t004:** Histological changes in quadriceps muscle induced by regular exercise training in rats supplemented with vitamin C and silymarin.

Group	Hypertrophy	Polygonal Fiber Shape	Striated Appearance	Endomysium	Epimysium	Perimysium	Vascularization	Splitting	Inflammation	Lipid Content	Cyanophile Sarcoplasm
CON	0	0	+	++	++	+++	0 a +	0	+	++	0
ET	++	+	++	+	+	++	+	+++	++	+	++
ET + VC	+++	++	++	+	+	++	+++	++	0/+	0/+	0
ET + SM	++++	++++	+++	+	++	+	++++	+	0	0/+	0

Criteria: 0 = Null (no presence/appearance). + = Low (minimal presence/appearance). ++ = Moderate (intermediate presence/appearance). +++ = High (important presence/appearance). ++++ = Very high (remarkable presence/appearance).

**Table 5 ijms-21-07724-t005:** Histological changes in the gastrocnemius muscle induced by regular exercise training in rats supplemented with vitamin C and silymarin.

Group	Hypertrophy	Polygonal Fiber Shape	Striated Appearance	Endomysium	Epimysium	Perimysium	Vascularization	Splitting	Inflammation	Lipid Content	Cyanophile Sarcoplasm	Satellite Cells	White Fibers	Red Fibers
CON	0	0	++	++	+++	++	+	0 a +	0	++	0	0 a +	+	+++
ET	+++	+++	+++	+	++	++	+++	++++	0 a +	+	++	0 a +	+	+++
ET+ VC	++++	+++	+++	0	+	++	+++	++	0	0	0	0	+++	+
ET + SM	++++	++++	++++	0	+	++	+++	+	0	0	0	0 a +	++	+++

Criteria: 0 = Null (no presence/appearance). + = Low (minimal presence/appearance). ++ = Moderate (intermediate presence/appearance). +++ = High (important presence/appearance). ++++ = Very high (remarkable presence/appearance).

**Table 6 ijms-21-07724-t006:** Histological changes in the myocardium induced by regular exercise training in rats supplemented with vitamin C and silymarin.

Group	Hypertrophy	Endocardium	Pericardium	Vascularization	Splitting	Eosinophilia
CON	0	0/+	+++	+	0/+	+
ET	+++	+	+	+++	++++	+++
ET + VC	+++	+	+	+++	0/+	+
ET + SM	+++	+	+	+++	0/+	+

Criteria: 0 = Null (no presence/appearance). + = Low (minimal presence/appearance). +++ = High (important presence/appearance). ++++ = Very high (remarkable presence/appearance).

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
