# Peer review of "Effect of Silymarin Supplementation on Physical Performance, Muscle and Myocardium Histological Changes, Bodyweight, and Food Consumption in Rats Subjected to Regular Exercise Training"

_ijms, 2020, doi:10.3390/ijms21207724_

Round 1
Reviewer 1 Report
This manuscfipt describes effect of silymarin on physical performance in rats subjected to regular exercise training.
Although it is extremely disappointing a single-dose in vivo study without the evaluation of dose-response, the referee would agree with this manuscript after consideration described below.
1. The authors need to explain closely why vitamin C was chosen as a reference compound. Quercetin, which having a more similar chemical structure to silymarin than that of vitamin C, has been reported by the related research cited in Ref. 29 and 30.
Author Response
Reviewer comment:
The authors need to explain closely why vitamin C was chosen as a reference compound. Quercetin, which having a more similar chemical structure to silymarin than that of vitamin C has been reported by the related research cited in Ref. 29 and 30.
Answer:
1. We appreciate your comment. We decided to use vitamin C as a reference because possess more scientific evidence as antioxidant in sports training protocols than quercetin. Quercetin is been recently use as ergogenic in sports but does not have yet enough evidence to use as a control. Even, International Society of Sports Nutrition at the recent document regard to “ISSN exercise & sports nutrition review update: research & recommendations” published on 2018, stated vitamin C in category I according to the classification of dietary supplements in sports, considering as follows:
1. Strong Evidence to Support Efficacy and Apparently Safe: Supplements that have a sound theoretical rationale with the majority of available research in relevant populations using appropriate dosing regimens demonstrating both its efficacy and safety.
Instead, quercetin is yet considering a supplement in category II. Regard to this category ISSN stays:
- Limited or Mixed Evidence to Support Efficacy: Supplements within this category are characterized as having a sound scientific rationale for its use, but the available research has failed to produce consistent outcomes supporting its efficacy. Routinely, these supplements require more research to be completed before researchers can begin to understand their impact. Importantly, these supplements have no available evidence to suggest they lack safety or should be viewed as harmful.
On the other hand, International Olympic Committee on its “IOC consensus statement: dietary supplements and the high-performance athlete” published in 2018, point out that vitamin C can be consider as a nutrient that may quenches reactive oxygen species reducing oxidative stress and augmenting immunity. Evidences suggest that reduces interleukin-6 and cortisol responses to exercise in humans. Besides, consider vitamin C as part of the supplements that may assist with training capacity, recovery, muscle soreness and injury management
That is why we consider trying with vitamin C instead of quercetin in this study; this additional information is found on lines 337-347 and 670-672.
References:
- Kerksick CM, Wilborn CD, Roberts MD, Smith-Ryan A, Kleiner SM, Jäger R, Collins R, Cooke M, Davis JN, Galvan E, Greenwood M, Lowery LM, Wildman R, Antonio J, Kreider RB. ISSN exercise & sports nutrition review update: research & recommendations. J Int Soc Sports Nutr. 2018 Aug 1;15(1):38. doi: 10.1186/s12970-018-0242-y. PMID: 30068354; PMCID: PMC6090881.
- Maughan RJ, Burke LM, Dvorak J, Larson-Meyer DE, Peeling P, Phillips SM, Rawson ES, Walsh NP, Garthe I, Geyer H, Meeusen R, van Loon LJC, Shirreffs SM, Spriet LL, Stuart M, Vernec A, Currell K, Ali VM, Budgett RG, Ljungqvist A, Mountjoy M, Pitsiladis YP, Soligard T, Erdener U, Engebretsen L. IOC consensus statement: dietary supplements and the high-performance athlete. Br J Sports Med. 2018 Apr;52(7):439-455. doi: 10.1136/bjsports-2018-099027. Epub 2018 Mar 14. PMID: 29540367; PMCID: PMC5867441.
We hope that our responses to the reviewers' observations are satisfactory and are available at any time for doubts and concerns related with this new version of our manuscript.
Sincerely yours,
Nancy Vargas-Mendoza, M Sc
José Antonio Morales, MD, PhD.
Reviewer 2 Report
The manuscript entitled “Effect of silymarin supplementation on physical performance, muscle and myocardium histological changes, body weight, and food consumption in rats subjected to regular exercise training” was attempted in a thorough manner. But the following comments need to be addressed before publishing in the journal.
COMMENTS
1. Figure 3, it is not complete from the figure given in the manuscript. Please check the figure once again for the correctness.
2. The interconnection between the parameters studied was given clearly and thoroughly.
3. The conclusion is too broadly given. A precise conclusion can be attractive and easy to understand by the readers once published.
4. The references need to be checked for the correctness according to the journal’s format as some of the references are not matching with other references. Please use the IJMS FORMAT for referencing.
Author Response
Reviewer comment:
The manuscript entitled “Effect of silymarin supplementation on physical, muscle and myocardium histological changes, body weight, and food consumption in rats subjected to regular exercise training” was attempted in a thorough manner. But the following comments need to be addressed before publishing in the journal.
1. Figure 3, it is not complete from the figure given in the manuscript. Please check the figure once again for the correctness.
- Answer.The figure 3 has been corrected as you request. The format of the figure was modified and the quality was improved for a better view and appearance. Thank you.
2. The interconnection between the parameters studied was given clearly and thoroughly.
- Answer. In fact, in this study we evaluate the effect of silymarin supplementation in rats exposed to a 8 week-regular exercise training program, we observed at the time that muscle quadriceps/gastrocnemius and myocardium tissues changed in the several characteristics measured, the physical performance was improved significantly in the maximal endurance capacity test. Also, we could stablish a clear interconnection with the other variables of body weight and food consumption in the sense than body weight silymarin group increased similar to control, however, the difference is that histological analyses showed apparent alteration in body composition augmenting muscle fiber hypertrophy and reducing fat content. This is evidently stated along the discussion in the manuscript especially on lines 596-610 and 626-631 in yellow color.
Thank you for your comment.
3. The conclusion is too broadly given. A precise conclusion can be attractive and easy to understand by the readers once published.
- Answer. The conclusion was delimited and adjusted considering the most relevant information resulted in this research. We decided to remove the text in lines 717 to 721 in order to give a better understanding and clear idea of this work to readers.
4. The references need to be checked for the correctness according to the journal’s format as some of the references are not matching with other references. Please use the IJMS FORMAT for referencing.
- Answer. The format for references has been corrected, we use IJMS FORMAT. Thank you for your comment.
As a general point, we wish to mention that the editing of the article in English has been carried out by a certificate English editor.
We hope that our responses to the reviewers' observations are satisfactory and are available at any time for doubts and concerns related with this new version of our manuscript.
Sincerely yours,
Nancy Vargas-Mendoza, M Sc
José Antonio Morales, MD, PhD.